# Disseminating implementation science: Describing the impact of animations shared via social media

Michael Sykes[1]*, Lucia Cerda[2], Juan Cerda[2], Tracy Finch[1]

**1** Department of Nursing, Midwifery and Health, Northumbria University, Newcastle Upon Tyne, United Kingdom, **2** Lucia Cerda Design, Barcelona, Spain

* Michael.sykes@northumbria.ac.uk

## Abstract

### Background

Dissemination is an under-researched activity that is important to researchers and funders and may have a role in creating conditions for implementation. We aim to study the impact of two animations shared via social media upon dissemination.

### Methods

We developed two short animations to increase the awareness of healthcare positional leaders of work undertaken to enhance a much-used implementation intervention. We measured both how frequently the related articles were accessed, and engagement with the research team, before and after the intervention. We analysed variation over time using statistical process control to identify both trend and periods of significant change.

### Results

We found evidence that the animation increased how frequently the articles were accessed, with a significant increase (p = <0.01) during the two weeks after release of each animation. One animation was associated with an increase in positional leaders' engagement with the research team.

### Conclusions

Animations shared via social media can enhance dissemination. We describe lessons from the work to develop the intervention and support calls for work to increase the understanding and adoption of effective dissemination interventions. Our findings provide support for further work using randomised study designs.

**Data Availability Statement:** All relevant data are within the paper and its Supporting Information files.

**Funding:** This report is independent research arising from a Doctoral Research Fellowship (DRF-

2016-09-028) supported by the National Institute for Health Research (NIHR). The views expressed in this presentation are those of the authors and not necessarily those of the NHS, the National Institute for Health Research or the Department of Health and Social Care. The funders had no role in study design, data collection and analysis, decision to publish, or preparation of the manuscript.

**Competing interests:** The authors have declared that no competing interests exist.

## Introduction

Dissemination of research helps towards meeting societal, funder and researcher goals: At the societal level, dissemination has a role in increasing research use across sectors such as health, social care, criminal justice and education, towards the goals of changing behaviour, increasing the quality of services and improving outcomes [1]. Dissemination may have this effect by addressing awareness, knowledge, perceptions, and motivation [2], leading to behaviour change [3–5]. Nilsen [6] describes a diffusion-dissemination-implementation continuum, where "dissemination is the active spread of new practices to the target audience using planned strategies" (p2). Such dissemination requires consideration of the recipients' needs, tailoring dissemination to cultural and structural features, using appropriate style, imagery, communication channels [7] and the settings in which research findings are to be received [8].

Funders expect publicly-funded researchers to undertake activities to disseminate their research to multiple research users as part of the work to develop impact [9–11]. Funders often provide resource to support dissemination of findings, and many provide structures to transmit the findings [12, 13]. Capturing the impact of dissemination activities is important to funders [14] and provides the opportunity for researchers to monitor and improve effectiveness of dissemination activities. The number of times a paper has been accessed can be used as a measure of 'user pull' uptake of research [15] and provides a measure of dissemination [16, 17].

Wilson et al. [8] found that dissemination was rated as important or very important by 93% (n = 216) of respondents to their survey of principal investigators of applied and public health research. The most common reported communication channel was through academic journals (98%) and conference presentations (96%). The use of animation or social media was not included in the list of options in their 2003–8 study, which pre-dated the commonplace use of social media, although 5% referred to using 'other' communication channels and one respondent described the use of a DVD.

Twitter is an important social media channel for communicating research findings: 9.4% of PubMed and Web of Science papers 2010–2012 had been tweeted at least once by the end of 2012 [18]. An emailed survey of people who tweet academic articles found that 47% (n = 856) sought to communicate to the public, 43% sought to communicate to peers [19]. A further survey of clinicians found they described using social media both to get research evidence (26.9%; n = 852) and to disseminate research evidence (15.0%) [20]. This supports earlier findings that the most identified reasons for health professionals social media use were extending colleague network of colleagues, updating colleagues about work and sharing information on medical conferences with my colleagues and marketing [21].

## Literature review

McGuire's Persuasive Communication Matrix [22] proposes five dissemination variables: the channel, source, message, audience, and setting. Social media is a valuable channel for dissemination and is associated with increased downloads and citations [23, 24]. Social media enables the use of more accessible content (for example, through using more visual presentation style [24]) and can provide information through a route that is more consistent with how clinical staff access information (for example, on a smartphone rather than through articles viewed on a hospital computer [25]). Twitter is the most commonly studied social media platform in correlational studies seeking to increase the impact of health research [26] and a valuable social media channel for health research [27]. There are conflicting findings about the effect of Tweets alone (without animation) upon downloads, attention (as measured by Altmetrics) or citations [28–30]. Adding infographics or podcasts [31] or a graphic summary of a research

article's question, methods, and major findings [32] to promotional tweets may increase article accesses.

Animation shared via social media, as part of health promotion campaigns to patients and the public, have reported positive outcomes including to reduce alcohol use [33] and increase awareness of neurosurgery [34] and of COVID [35]. We have not been able to identify previous work using animation shared via social media to target healthcare professionals. Animations have, however, been used as part of educational interventions targeting healthcare workers to improve clinical assessment [36], to improve pain management [37] and to improve nurses' response to cardiac arrest [38]. A pilot study by Attin and colleagues [38] randomised nursing students to receipt of an animation and discussion and reported faster responses to a cardiac arrest compared to a control group. A before-and-after study [39] delivered an animation to in-patients and nurses; it found that over 65 year olds fell significantly less after delivery of the animation, although the effect was not seen in patients younger than 65. Interestingly, there were no reported differences in knowledge, suggesting that this may not be the mechanism through which the animations had an effect.

There have been calls for research that extends dissemination science and practice [2], including the effectiveness of dissemination activities at achieving defined goals [7]. The effectiveness of dissemination activies might be influenced by various factors, these include the skills of those developing the dissemination materials, method of delivery and the wider context; as such, dissemination activies are complex interventions [6]. Guidance recommends that those developing complex interventions incorporate evidence, theory and stakeholder views; consider implementation and use iterative design methods [40, 41].

The current paper describes work to disseminate research findings from two studies to positional leaders in healthcare through animations shared via social media (Twitter). The source was an applied health researcher, describing work undertaken as part of a PhD to describe and enhance audit and feedback. The work was undertaken with a supervisory team, and involved substantial stakeholder involvement through co-production and advisory groups [42, 43]. The message of the first animation was a multisite description of what currently happens when a national audit reaches the hospital and how it could be enhanced through an intervention to support recipients to analyse performance, select strategies and generate commitment. The message of the second animation was a multisite description of what happens during an audit of the quality of care on wards across a hospital. Both messages were delivered through the medium of animation, a form of non-text output [44] and time-based visual artwork that illustrates a story unfolding over time [45]. The target audience for both animations was positional healthcare leaders (for example, directors, clinical audit leads). It was anticipated that they would be viewing social media in work, home or travel settings. In this paper, we describe the effect of the animation upon the frequency with which particular open-access papers [42, 43] were accessed.

## Materials and methods

### Aim

To study the impact of two animations shared via social media upon dissemination.

### Study design and theoretical framework

This is a phase II [46] before-and-after study using statistical process control to describe the impact of an animation intervention upon the weekly number of accesses of the target articles. We conceptualise dissemination as a process, an outcome of which is whether an article was subsequently accessed. We propose that whether the article is cited or acted upon is influenced

by the article content, rather than the dissemination activity. We recognise that the accessing of articles will vary over time (e.g. due to holiday periods); by monitoring the significance of variation we are able to describe the impact of the animation as an event in the dissemination process [47].

## Setting and participants

This was an online study where participants were Twitter users. The source Twitter account had approximately 530 followers, with further participants able to view the tweet as a result of likes, retweets and searches. The followers were predominantly described in their biographies as healthcare improvement leaders, clinicians, clinical positional leaders, clinical academics and people interested in implementation science.

The study was approved by the Newcastle University Ethics Committee (Ref: 3917/2020).

## Intervention

The animations, which are described in Box 1 and in the TIDieR checklist (S1 Checklist), are available at: https://twitter.com/Msykes09/status/1330556748210515968?s=20 and https://twitter.com/Msykes09/status/1408675290176491521?s=20.

Intervention development involved initial scoping of the content to be delivered, consideration of the message, the audience, the channel and the style, followed by iterative stakeholder

---

### Box 1. A description of the animation content and delivery

**The source:** An applied health researcher, describing work undertaken as part of a PhD to describe and enhance audit and feedback in dementia care. The work was undertaken with a supervisory team, and involved substantial stakeholder involvement through co-production and advisory groups [25, 26].

**The channel:** Twitter account with approximately 530 followers.

**Article 1 message:** A description of what currently happens when a national audit reaches the hospital and how it could be enhanced through an intervention to support recipients to analyse performance and select strategies.

**Article 2 message:** A description of what currently happens during a nurse-led hospital-wide audit of the quality of care in wards and steps to enhance the audit.

**The audience:** Positional healthcare leaders (for example, directors, clinical leads). For article 1, they were targeted through a Tweet directed to regional and national organisations leading healthcare improvement (e.g. @TheIHI @FabNHSStuff @Improvement-Cym @Aqua_NHS @Improve_Academy), national improvement and regulation agencies (@HIQA @HSCQI @noca_irl @online_his) and an umbrella organisation for English hospital providers (@NHSConfed). For article 2, they were targeted through Tweets directed to national nursing leads (e.g. @CNOEngland, @CharlotteMcArdl), directors of nursing and nursing research leaders (e.g. @AlisonProf, @DrJoanne_Cooper).

**The setting:** It was anticipated that they would be viewing social media on a phone or tablet in work, home or travel settings.

---

engagement and refinement (Fig 1). Stakeholder engagement resulted in refinements: to increase appropriateness by extending the diversity in the images to more closely reflect the target audience; to increase accessibility by abridging the text, changing the duration of each scene and simplifying the language (e.g. changing 'sites' to 'hospitals'); and to increase engagement and recipient sensemaking by amending the language and adding the learning outcomes to the beginning of the animation.

The animations were developed to be engaging: They included content to support the viewer to compare the findings to their own practice, by describing current practice and asking whether this "sounds like what happens at your hospital"; They included content relating the work to priorities, by presenting benefits for patients, clinicians and positional leaders. The animations used accessible language and employed a positive, improvement-focussed tone intended to be acceptable to recipients. It was anticipated that describing the funder, university and stakeholder involvement would develop perceived credibility and trustworthiness. The animations were produced using Adobe Illustrator and After Effects.

The animations were released at different times of the year, with the link to the article in a linked tweet and the first author's (MS) biography (Table 1).

## Data collection

The primary and target behavioural outcome of the animation was the number of times the paper had been accessed. This was retrieved from the target journals' website. We captured events that may confound the results (e.g. conferences, newsletter or non-study tweets). The secondary outcome was direct requests to the first author (MS) for further information. For Article 2, we recorded how often the video was viewed each week (S1 Table).

## Analysis

Statistical process control (SPC) charts describe system performance over time. We used SPC charts to describe the weekly number of accesses of the target papers before and after the animation. In line with guidance [30], we labelled the SPC with events in the dissemination process (potential confounders), for example, when the work described in article 1 was presented by the research team at an international conference, or when article 2 was highlighted in a newsletter independent of the research team.

The weekly number of accesses was analysed using an Excel-based statistical process control c-chart tool [30].

$$\bar{c} = \sum_i^m = 1\, x_i$$

Control limits = $\bar{c} \pm 3\sqrt{\bar{c}}$

The trend in the first three weeks after publication demonstrated special cause variation and, in line with guidance for the use of SPC, was removed from the graph [48]. The SPC charts describe the subsequent trend for the period before and after the release of the animations. Visual analysis of the SPC charts sought temporal association between the intervention and the primary outcome (article accesses) and secondary outcome (contacts to corresponding author).

## Results

Statistical process control charts are interpreted by looking for occasions when the frequency line crosses the standard deviation line. Crossing the three standard deviation line is evidence for significant difference (p = <0.01), and is referred to as special cause variation. Remaining

First author (MS) summarised the target articles to the second (JC) and third authors (LC) who asked exploratory questions in relation to the audience and key messages for audience sub-groups.

Rationale: To identify the content and delivery by specifying:

- target audience (people with control over improvement),

- delivery mode (via Twitter),

- implementation outcomes (to be acceptable, accessible and engaging)

- content to achieve implementation outcomes (the animation 'tone' should be descriptive, non-critical and offer solutions and the language accessible). To increase engagement through the illustration of scenes familiar to the target audience (e.g. bedside, office, committee room), the careful use of movement, paying attention to complexity (the cognitive load) and describing the source of the information, which was anticipated to be credible.

Output: Following initial discussion between MS, JC and LC, MS produced a brief for the content of the two animations (Appendix A)

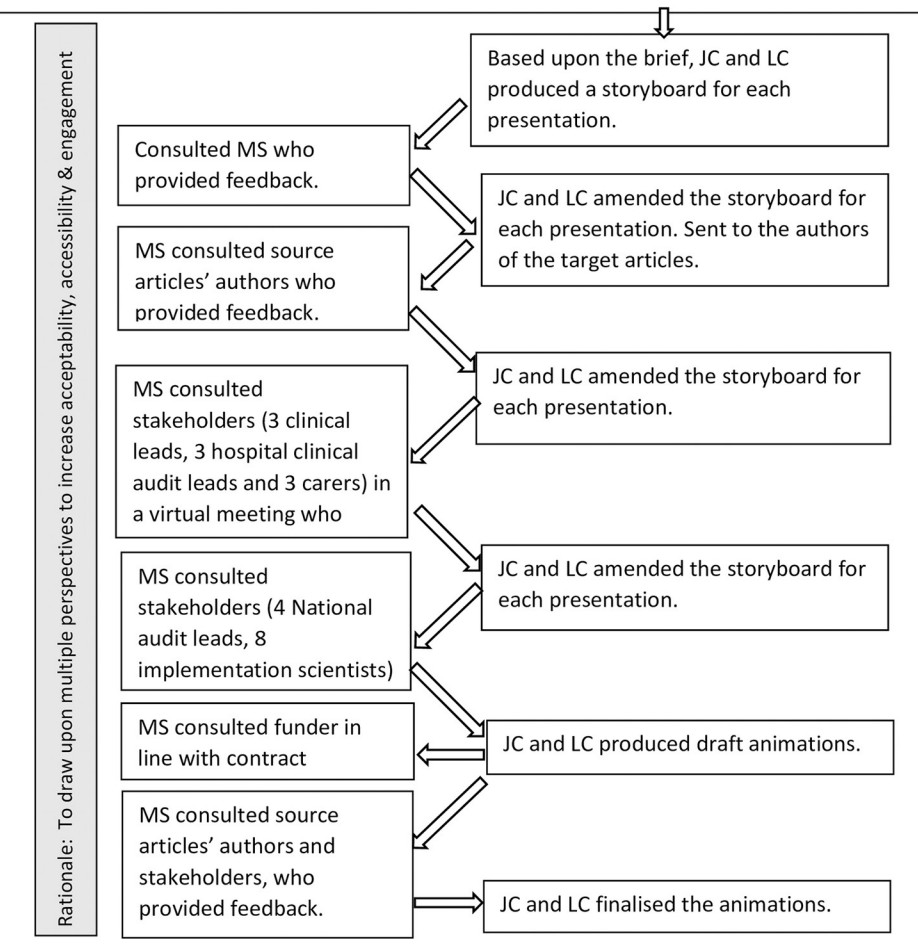

**Fig 1. A flowchart describing intervention development.**

**Table 1. The source, date and content of the Tweets delivering the animations.**

| | |
|---|---|
| Animation 1 was tweeted by the first author (MS) on 17th November 2020 | *Here's a short animation of our @NIHRresearch funded work to describe and enhance a national audit. The work was undertaken with @TracyLFinch @niinamk @drlouiseallan and Richard Thomson #NUPHSI. It would not have been possible without the input from a large number of stakeholders, including @afMetaLab and many others whom I am unable to name here.* |
| Animation 1 was tweeted by MS on 22nd November 2020 | *Enhancing what happens when a national audit reaches the hospital. May be of interest to @THIS_Institute @TheIHI @FabNHSStuff @HIQA @HSCQI @noca_irl @online_his @ImprovementCym @Aqua_NHS @Improve_Academy @FNightingaleF @NHSConfed* |
| Animation 1 was tweeted by MS on 23rd November 2020 | *Those following #CAAW20 may be interested in this.* Note: Using #CAAW20 sought to bring it to the attention of people following the Clinical Audit Awareness Week |
| Animation 2 was tweeted by MS on tweeted on 26th June 2021 | *NEW animation: Describing a monthly audit of ward quality at NHS [National Health Service] hospitals.* The thread continued: *You can read more about what we did, what we found and what we propose here:* [Link to paper] |
| Animation 2 was tweeted by MS on 2nd July 2021: | *Directors of nursing, matrons + ward managers described ward audit data as 'meaningless' & 'a sea of green'. @DavidFMelia @RivkahMiar @Antonialynch @mapFlynn @AlisonSmith2306 The animation & paper describe that the audit is costly, may have adverse effects & could be enhanced @Karen_Goudie @Day2H @lzredfernsoecno @KarenDunderdale @LeesLizzie @angelawooduk @PeteWRN* |
| Animation 2 was tweeted by MS on 4th July 2021: | *The paper and animation describe the discomforting issue of punitive feedback and its impacts upon patient care, staff wellbeing, improvement, assurance and cost.* |
| Animation 2 was tweeted by MS on 13th July 2021: | *A multi-site study using interviews, observations & doc analysis, supported by 2 groups of stakeholders. Funded by @NIHRresearch Found opportunities to improve care, staff well-being, costs & assurance. @AlisonProf @DrJoanne_Cooper @jorycroftmalone @PorteousDr @Evidence4QI* |

within the standard deviation line is evidence that whilst there might be variation, this is not significantly different from prior frequency and is referred to as common cause variation.

The mean number of weekly accesses for article 1 was 25, with a significant decrease evident during the last week of August. In the two weeks after the intervention, there was a significant increase in the weekly number of accesses, as described in Fig 2. There was common cause variation (that is, no significant change) after presenting the paper at an international conference or by tweeting links to the paper without the animation during weeks 2 and 13.

The mean number of weekly accesses for article 2 was 43, with a significant decrease evident during the third week of July. In the two weeks after the intervention, there was a significant increase in the weekly number of accesses, as described in Fig 3. Article 2 also had significant increases in the number of accesses following publicity about the paper by @Evidence4QI (a project seeking to implement evidence in quality improvement projects) and after posting in the Q Community (https://q.health.org.uk/) discussion forum. Whilst study tweets were re-tweeted; we sought but did not identify novel non-study tweets.

The secondary outcome was direct requests to the first author for further information. We received seven requests for further information about article 1 during the study period. Five of the requests came during the 8 days after the animation was released (Table 2).

In addition, we received five requests for further information about article 2 during the study period, one of which, to speak to an improvement lead from a UK healthcare provider organisation, came in the period immediately after the release of the animation (Contact received on 26h June). The other four came later and shortly after sharing the animation via a

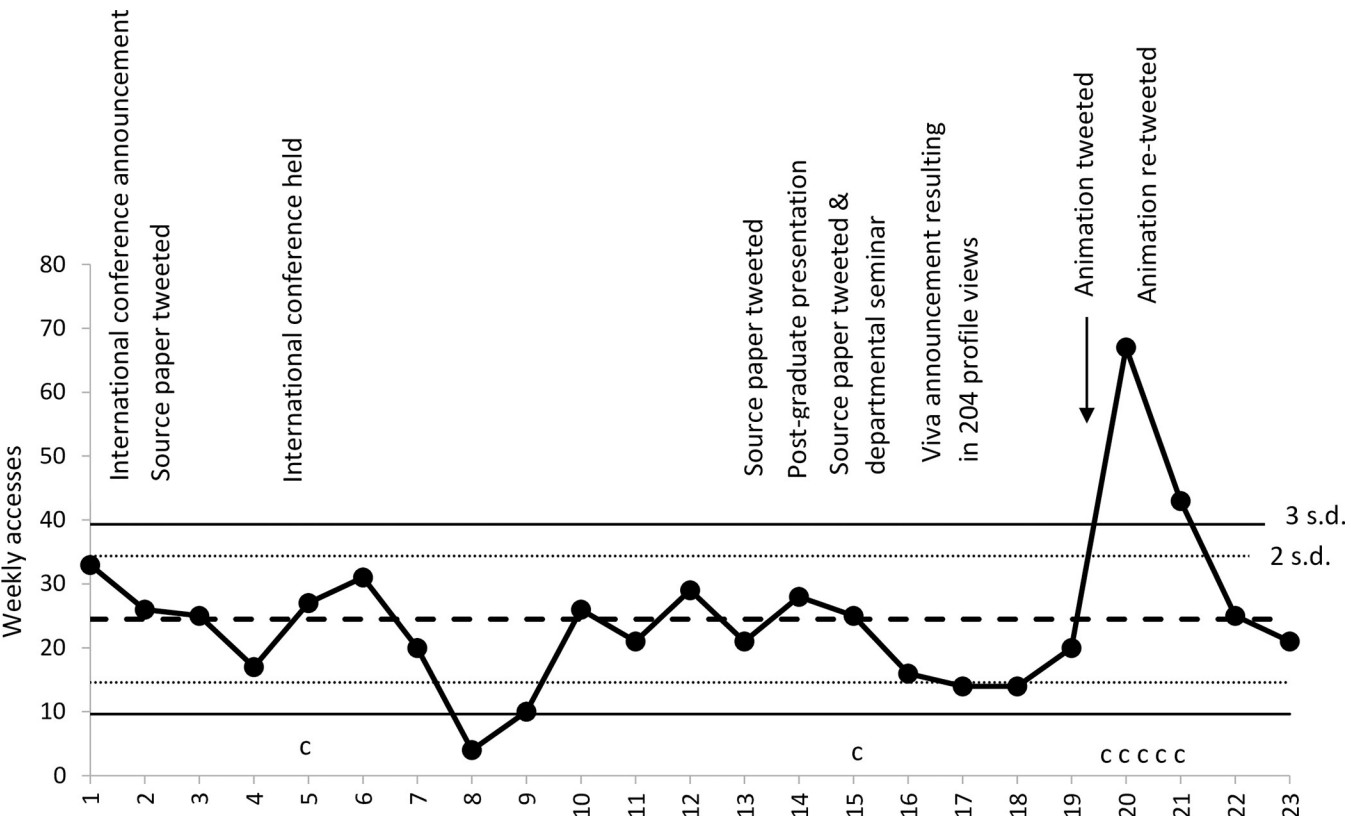

**Fig 2. A statistical process control c-chart showing weekly accesses of Sykes et al.** (2020) and numbers of contacts [c] made to first author by stakeholders [s. d. = Standard deviation].

quality improvement virtual discussion forum: National leads 4th and 10th August; Organisational leads 5th and 10th August.

## Discussion

There is evidence that the animation intervention initiated the intended response in the target audience. We found that each animation was associated with a significant increase in the weekly number of accesses. Specifically, during the two weeks after the intervention there was a significant increase. Over these two weeks, there were 60 accesses above the mean for the proceeding period for article 1 and 24 accesses above the mean for the proceeding period for article 2. There is evidence that the animation 1 was temporally associated with new requests for information and discussion from the target audience.

There are strengths and limitations to the work: Consistent with McGuire's Persuasive Communication Matrix, we identified the source, channel, message, audience and setting [22]. The animations were designed to meet a specific outcome (for viewers to be motivated to gain more information about the research described in the animation) and to meet specific dissemination outcomes (to be engaging, accessible and acceptable). The iterative, multi-method stakeholder engagement is a further strength. The overall effect of the engagement was to support the use of animations delivered via Twitter, and both to refine delivery and focus the message so that messages perceived as being more peripheral were removed. This highlights the role of stakeholder engagement in identifying the messages to be disseminated.

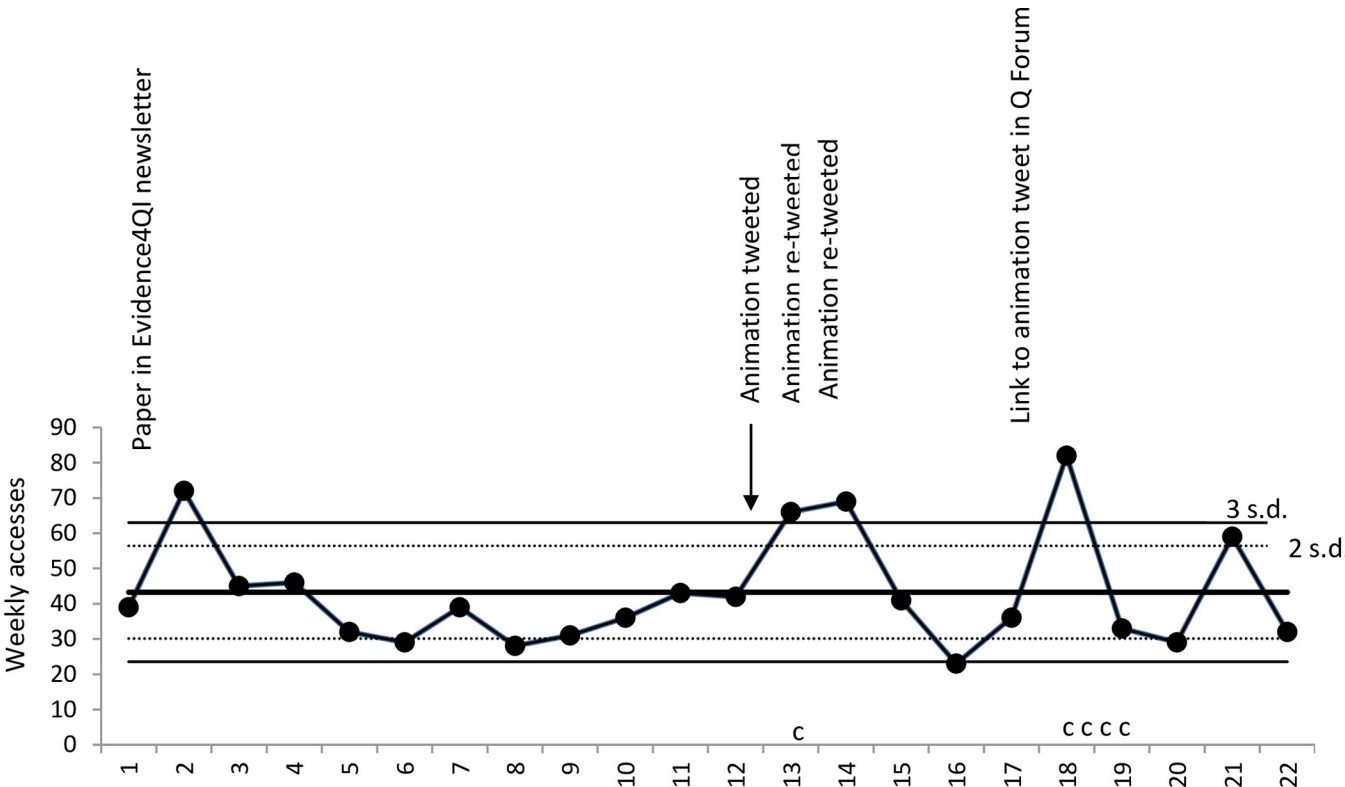

**Fig 3. A statistical process control c-chart showing weekly accesses of Sykes et al.** (2021) and numbers of contacts [c] made to first author by stakeholders.

It is possible that the findings are not generalisable to different animations, articles, sources or audiences. Article 1 was published in an open-access journal (Implementation Science). Implementation Science is the third ranked health policy journal [49] with a 2-year impact factor of 4.525 [50]. Article 2 was published in an open-access journal (BMC Health Services Research), the 66th ranked health policy journal [49] with a 2-year impact factor of 2.655 [51]. It is possible that accesses would have been reduced if access were behind a paywall, or if published in a different journal. The animations were disseminated from a personal account. It is possible that the impact of the animations was affected by the source; future work should consider the influence of changing the source, for example, to an organisational account. Due to Twitters limited number of characters, the associated papers and the inclusion of target recipients were tweeted as a reply to the animation, thereby requiring an additional action to view the animation and to open the paper; it is possible that this acted as a barrier to accessing the

**Table 2. A table to show when requests for further information about article 1 were received.**

| Date (2020) | Contact to first author |
| --- | --- |
| 27th July | Request to speak to a deputy director of nursing about the study |
| 13th October | Request to speak to a national clinical lead about the work |
| 19th November | Invite to speak to a Trust-level committee |
| 21st November | Invite to speak to a national network of clinical audit leads |
| | Contact from the clinical lead of a national audit |
| 24th November | Invite to speak to a Trust-level committee |
| 25th November | Request to use the animation in a presentation by a non-executive to a Trust-level committee |

article. Future work could explore the impact of targetting individuals through the inclusion of Twitter handles. We did not collect data on impressions, engagements, details expands or profile visits, and only collected views for article 2. It is possible that this would have provided a richer picture of how recipients responded to viewing the animation. Data collection was extracted manually from the journal website; like previous studies [17] we are not able to confirm the accuracy of the count. During data collection for the second animation, the counter did not change during the period 2nd to 5th July (cumulative 871 accesses) and 6th to 13th July (cumulative 882 accesses). These were the only multi-day periods when the journal counter recorded that the articles were not accessed; this hiatus was followed by a large increase (88 accesses). It is anticipated that this was an error in the data collection by the journal. Email correspondence with the journal sought, unsuccessfully, to correct this. As a result, the data analysis included an estimate for data distribution for this two week period based upon article 1. For transparency, the uncorrected data is presented in S1 Fig and S1 Table. Both the corrected and uncorrected graphs illustrate a significant increase in accesses in two of the three weeks after the release of the second article. Consistent with the guidance [48, 52], we re-calculated the mean and standard deviation after the initial period of special cause variation.

The current study focussed on usual dissemination and the social media delivery of animations targeted at positional leader knowledge users. The main outcome was for recipients to seek further information, as measured through article access. It cannot be inferred that those accessing the paper where the target audience or that they behaved differently as a result of accessing the article. However, subsequent contacts were from members of the target audience. The animations cost £508; we did not monitor time costs, but these were estimated to be 25 hours of first author time. The animation intervention was associated with a significant increase in accesses that lasted two weeks. Future work should consider how to enhance the response, for example, a multi-faceted approach may result in a more sustained increase [53]. Future research should also consider the link between increased article accessed and author contact, and impact upon target audience behaviour and patient outcomes.

There is a diverse range of dissemination methods that vary by complexity and level of interaction [54]; for example, Coon et al. [55] describe creative communication methods associated with systematic reviews, including methods for use in dissemination (illustrations, podcasts, blog posts, briefing papers, board games, social media shareable content). Video abstracts, which used the plain language summaries as the spoken script, can lead to greater comprehension, reported understanding and a positive affective response than original or graphic abstracts [56]. Our study extends this work by finding that they may also increase engagement with research articles and researchers.

There were important lessons from developing the animations. The mode of delivery impacted upon number and content of the scenes: there is a 140 second limit to embedded videos on Twitter and we found there was a minimum length of time needed to understand a scene. This constrained the number and cognitive load of scenes, including the volume of text and use of movement. In developing the animation, the role of movement was a key consideration: we sought to balance the use of movement to gain attention with the impact of movement on cognitive load that could distract from participant engagement with the message. The animation was anticipated to be viewed on tablets and mobile phones, which impacted upon the size of the font. It was anticipated that this could be viewed in work, public or home settings, as such sound needed to be optional. Providing both text and voice may have added to the cognitive load. As a result, we opted for gentle music consistent with the intended tone rather than narration. The feedback from the consultation with stakeholders highlighted

opportunities to increase accessibility by improving the clarity about the research findings and addressing international differences in how audiences may view the findings.

## Conclusion

We studied the impact of animations to increase the frequency with which two open-access papers [42, 43] were accessed. In designing the intervention, we drew upon McGuire's Persuasive Communication Matrix [22] to consider the source, channel, message, audience, and setting. We found that the release of the animation was associated with a significant increase in the number of times the article was accessed. We observed that neither tweeting article links without the animation nor presentation at an international conference led to a significant increase. We propose that the use of animation distributed via Twitter may provide an effective way to disseminate research findings and increase stakeholder engagement with study findings. Further work to test the use of animation using randomised study design, and to investigate the impact upon dissemination outcomes from evidence newsletters and delivering animations via professional digital forums, would be valuable.

## Supporting information

**S1 Checklist. The TIDieR (Template for Intervention Description and Replication) Checklist**∗**.**
(DOCX)

**S1 File. The brief developed by MS, LC and JC.**
(DOCX)

**S1 Fig. The unadjusted statistical process control c-chart showing weekly accesses of Sykes et al. (2021).**
(DOCX)

**S1 Table. Weekly data per article.**
(DOCX)

## Acknowledgments

We would like to acknowledge the valuable input from stakeholders and advisors, in particular Anne Sales for early discussions about the work and feedback about the draft animation.

## Author Contributions

**Conceptualization:** Michael Sykes.

**Data curation:** Michael Sykes.

**Formal analysis:** Michael Sykes.

**Funding acquisition:** Michael Sykes.

**Investigation:** Michael Sykes.

**Methodology:** Michael Sykes.

**Project administration:** Michael Sykes.

**Resources:** Lucia Cerda, Juan Cerda.

**Supervision:** Tracy Finch.

**Visualization:** Lucia Cerda, Juan Cerda.

**Writing – original draft:** Michael Sykes.

**Writing – review & editing:** Michael Sykes, Lucia Cerda, Juan Cerda, Tracy Finch.

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
