## [Decision Letter · Decision Letter 0]

22 Apr 2022

PONE-D-22-05141Disseminating implementation science: Describing the impact of social-media delivered animationsPLOS ONE

Dear Dr. Sykes,

Thank you for submitting your manuscript to PLOS ONE. After careful consideration, we feel that it has merit but does not fully meet PLOS ONE’s publication criteria as it currently stands. Therefore, we invite you to submit a revised version of the manuscript that addresses the points raised during the review process.

This work has clear value to the scientific community, but needs a stronger background section, more detail, data and depth to achieve publication. The standard of English / writing is acceptable.==============================

We look forward to receiving your revised manuscript.

Kind regards,

Christopher Carroll, Ph.D.

Academic Editor

PLOS ONE

Journal Requirements:

“This report is independent research arising from a Doctoral Research Fellowship (DRF-2016-09-028) supported by the National Institute for Health Research (NIHR). The views expressed in this presentation are those of the authors and not necessarily those of the NHS, the National Institute for Health Research or the Department of Health and Social Care.”

Reviewers' comments:

Reviewer's Responses to Questions

**Comments to the Author**

1. Is the manuscript technically sound, and do the data support the conclusions?

Reviewer #1: Yes

Reviewer #2: Yes

2. Has the statistical analysis been performed appropriately and rigorously? 

Reviewer #1: Yes

Reviewer #2: I Don't Know

3. Have the authors made all data underlying the findings in their manuscript fully available?

Reviewer #1: Yes

Reviewer #2: No

4. Is the manuscript presented in an intelligible fashion and written in standard English?

Reviewer #1: Yes

Reviewer #2: No

5. Review Comments to the Author

Reviewer #1: Thank you for your submission, I found it most interesting. I have left various comments attached to the PDF which largely should be easy to attend. There are some early issues around formatting of references and there are a couple of areas that need rewriting for clarification. I also would have liked to have seen more data around the Tweet analytics that went beyond the publication views, so for example Tweet impressions and engagements as well as how many times the embedded videos were viewed over the same time you explored article access. Also, it is interesting to note that the publication was not Tweeted as part of the animation, but as a reply in a thread. It would be interesting to know why you took that approach and used the @ to gain attention in the animation but not as a subsequent Tweet. It is also important to note that the push for increased visibility will have potentially increased views of the paper if it was in the first Tweet rather than not include the handles of notable users, adding them to a reply. A comparison between different types of Tweets (with handles and animations as opposed to just article URL) and their impact on views would be useful for comparison and future exploration. This can be covered in the Discussion. It is a useful paper, although I do sometimes wonder whether these kinds of exercises are better served as a blog post than academic paper. That said, there is broad appeal for this kind of work and it will be of interest to those outside of your discipline.

Good luck with your amendments and going forward.

Reviewer #2: Thanks for the opportunity to review this paper. This study addresses an important question in academia with an innovative way. The research findings are valuable to the literature and the practice. To enhance this paper for publication, I would like to suggest the following revisions:

1. It would be helpful for readers if the introduction and literature review are put into two separate sections. The first three paragraphs could be the introduction and the rest paragraphs could be the literature review.

2. In the introduction, more discussion about why dissemination is important is needed. The authors touched on this in the fourth paragraph (“Dissemination activity may address antecedents of behaviour change, including awareness, knowledge, perceptions, and motivation [12].”). But more detailed discussion is preferred.

3. The literature review discussed social media as a useful tool for research dissemination. It would be better if the authors provide more examples about how social media have been used for research dissemination. There are a bunch of survey studies investigating how health researchers and professionals use social media to disseminate and learn about trending research.

4. A review of literature on using animations to enhance dissemination should be added to the literature review section. If there is little research on using animations for research dissemination, the authors could review literature about using animations to enhance dissemination of other types of messages (e.g., health messages, news reports, etc.). Also, formal hypotheses should be proposed following the literature review.

5. Moving to the method section, the details about animation development is appreciated. But it would be more clear to readers if the process is summarized and presented with a flow chart. Also, the purpose of each step and the and justification of each decision should be clarified. For example, what is the role of the co-production group and why the authors decided to make the animations "descriptive, non-critical and offer solutions"? In addition, it would be better to replace the name initials with first, second, and third author. It is somewhat confusing to understand what MS, JC, and LC refer to when reading the paragraph.

6. The list of animation tweets could be summarized in a table.

7. The discussion about the generalizability should also consider the influence of source. The animation shared by different sources (e.g., organizational vs. individual researcher, source affiliation, source expertise, etc.) may have different impact on dissemination. With the single source in this study, the generalizability of the finding is undermined.

8. As a reader, I would also like to see more discussion of using animation in comparison with other possible media formats to enhance dissemination (e.g., podcast interview, infographics).

9. The appendix numbering should be checked and fixed. The A, B, C numbering system and the 1, 2, 3 numbering system are used at the same time.

10. I would suggest deleting appendix B because the study is not a randomized control trial.

11. Some of the sentences are difficult to understand due to complicated sentence structures. Please edit the manuscript to enhance the readability.

6. PLOS authors have the option to publish the peer review history of their article (what does this mean?). If published, this will include your full peer review and any attached files.

Reviewer #1: **Yes: **Andy Tattersall

Reviewer #2: No

---

## [Author Response · Author response to Decision Letter 0]

26 May 2022

Response to review feedback: Disseminating implementation science: Describing the impact of social-media delivered animations [PONE-D-22-05141]

Feedback 1: Please ensure that your manuscript meets PLOS ONE's style requirements, including those for file naming. The PLOS ONE style templates can be found at

Response 1: Apologies, we have amended the manuscript in line with the formatting template.

Feedback 2. Thank you for stating the following financial disclosure:

“This report is independent research arising from a Doctoral Research Fellowship (DRF-2016-09-028) supported by the National Institute for Health Research (NIHR). The views expressed in this presentation are those of the authors and not necessarily those of the NHS, the National Institute for Health Research or the Department of Health and Social Care.”

Response 2: We can confirm that the funders had no role in study design, data collection and analysis, decision to publish, or preparation of the manuscript. (Line 349)

Feedback 3. We note that you have stated that you will provide repository information for your data at acceptance. Should your manuscript be accepted for publication, we will hold it until you provide the relevant accession numbers or DOIs necessary to access your data. If you wish to make changes to your Data Availability statement, please describe these changes in your cover letter and we will update your Data Availability statement to reflect the information you provide.

Response 3: All available data is provided in the manuscript in graphical form and the new S3.

Feedback 4. Your ethics statement should only appear in the Methods section of your manuscript. If your ethics statement is written in any section besides the Methods, please move it to the Methods section and delete it from any other section. Please ensure that your ethics statement is included in your manuscript, as the ethics statement entered into the online submission form will not be published alongside your manuscript.

Response 4: We have added the ethics statement to the method. (Line 197)

Reviewers' comments:

Feedback a1. Is the manuscript technically sound, and do the data support the conclusions?

Reviewer #1: Yes

Reviewer #2: Yes

2. Has the statistical analysis been performed appropriately and rigorously? 

Reviewer #1: Yes

Reviewer #2: I Don't Know

3. Have the authors made all data underlying the findings in their manuscript fully available?

Reviewer #1: Yes

Reviewer #2: No

Response 3: All available data is now provided in the manuscript.

4. Is the manuscript presented in an intelligible fashion and written in standard English?

Reviewer #1: Yes

Reviewer #2: No

Response 5: We have amended the manuscript to increase clarity, as detailed below.

5. Review Comments to the Author

Reviewer #1: Thank you for your submission, I found it most interesting. I have left various comments attached to the PDF which largely should be easy to attend. There are some early issues around formatting of references and there are a couple of areas that need rewriting for clarification.

Response 1.1: We are extremely grateful to the reviewer for this approach, and have amended the document in line with their comments:

• Change to ‘animation shared via social media’ (Lines 4, 24, 34, 90, 93, 112, 129)

• Added additional keywords

• Removed e.g. from references

• Clarified that Wilson study largely pre-dates commonplace social media usage (Line 65)

• Amended sentence describing the effectiveness of dissemination activities, so as to use the suggested wording (Line 106)

• Removed speech marks where not a quote

• Added study finding the tweets alone can be effective (Line 86)

• Clarified that MS refers to the first author on first usage within the manuscript (Line 169)

• Changed description of intervention development to a flowchart (Line 157)

• Described the animation production platform (Line 167)

• Added supplemental data of animation views to the Appendix (S3)

• That tweeting the paper as a reply may have impact upon engagement (Line 265)

• We have amended the discussion to note, “We did not collect data on impressions, engagements, details expands or profile visits, and only collected views for article 2. It is possible that this would have provided a richer picture of how recipients responded to viewing the animation.” (Line 267)

• I regret that we are not able to state the time that they were tweeted, as we did not record this information

• Included details of all contacts with the first author for article 2 (Line 230).

2. I also would have liked to have seen more data around the Tweet analytics that went beyond the publication views, so for example Tweet impressions and engagements as well as how many times the embedded videos were viewed over the same time you explored article access. 

Response 1.2: We agree that this could have provided greater insight into how recipients responded to the animation, and have included in the discussion that, “We did not collect data on impressions, engagements, details expands or profile visits, and only collected views for article 2. It is possible that this would have provided a richer picture of how recipients responded to viewing the animation.” (Line 268)

3. Also, it is interesting to note that the publication was not Tweeted as part of the animation, but as a reply in a thread. It would be interesting to know why you took that approach and used the @ to gain attention in the animation but not as a subsequent Tweet. It is also important to note that the push for increased visibility will have potentially increased views of the paper if it was in the first Tweet rather than not include the handles of notable users, adding them to a reply. 

Response 1.3: We agree with the reviewer and have amended the discussion to state that, “Due to Twitters limited number of characters, the associated papers and the inclusion of target recipients were tweeted as a reply to the animation, thereby requiring an additional action to view the animation and to open the paper; it is possible that this acted as a barrier to accessing the article”. (Line 263)

4. A comparison between different types of Tweets (with handles and animations as opposed to just article URL) and their impact on views would be useful for comparison and future exploration. This can be covered in the Discussion. 

Response 1.4: We agree and have included a statement that, “Future work could explore the impact of targeting individuals through the inclusion of Twitter handles.” (Line 267)

5. It is a useful paper, although I do sometimes wonder whether these kinds of exercises are better served as a blog post than academic paper. That said, there is broad appeal for this kind of work and it will be of interest to those outside of your discipline. Good luck with your amendments and going forward.

Response 1.5: We would like thank the reviewer for agreeing that there is a broad appeal to the work. Our view is that dissemination is an under-explored facet of implementation science, and that such peer-reviewed work will serve to ensure the quality of learning about interventions to increase reach and impact.

Reviewer #2: Thanks for the opportunity to review this paper. This study addresses an important question in academia with an innovative way. The research findings are valuable to the literature and the practice. To enhance this paper for publication, I would like to suggest the following revisions:

1. It would be helpful for readers if the introduction and literature review are put into two separate sections. The first three paragraphs could be the introduction and the rest paragraphs could be the literature review.

Response 2.1: We would like to thank the reviewer for this feedback. We have amended the introduction to reflect the two sections, whilst remaining consistent with journal format requirements. (Line 76)

2. In the introduction, more discussion about why dissemination is important is needed. The authors touched on this in the fourth paragraph (“Dissemination activity may address antecedents of behaviour change, including awareness, knowledge, perceptions, and motivation [12].”). But more detailed discussion is preferred.

Response 2.2: We agree and have extended the introduction to describe how dissemination helps towards meeting societal, funder and researcher goals. (Line 41-67)

3. The literature review discussed social media as a useful tool for research dissemination. It would be better if the authors provide more examples about how social media have been used for research dissemination. There are a bunch of survey studies investigating how health researchers and professionals use social media to disseminate and learn about trending research.

Response 2.3: We agree and have included a new paragraph describing Twitter as an important social media channel for communicating research findings. (Line 67)

4. A review of literature on using animations to enhance dissemination should be added to the literature review section. If there is little research on using animations for research dissemination, the authors could review literature about using animations to enhance dissemination of other types of messages (e.g., health messages, news reports, etc.). Also, formal hypotheses should be proposed following the literature review.

Response 2.4: We have included a brief review describing the use of animations as an educational tool and noting that we have not been able to identify studies exploring the use of animation delivered through social media targeting healthcare professionals. (Line 92)

5. Moving to the method section, the details about animation development is appreciated. But it would be more clear to readers if the process is summarized and presented with a flow chart. Also, the purpose of each step and the and justification of each decision should be clarified. For example, what is the role of the co-production group and why the authors decided to make the animations "descriptive, non-critical and offer solutions"? In addition, it would be better to replace the name initials with first, second, and third author. It is somewhat confusing to understand what MS, JC, and LC refer to when reading the paragraph.

Response 2.5: We agree and have added a flowchart illustrating intervention development. We have incorporated into the flowchart the rationale for each step and into the narrative before the flowchart the justification for decisions. I note that we tried re-writing MS, JC and LC with first second and third author, but this became repetitive and disrupted the flow. Instead, we have sought to increase clarity by using “first author (MS)”, “second author (JC)” and “third author (LC)” on the first occasion, then using initials. (Fig 1)

6. The list of animation tweets could be summarized in a table.

Response 2.6: We agree and have included them in a table. We have also tabulated the requests for further information. (Line 170, Table 1)

7. The discussion about the generalizability should also consider the influence of source. The animation shared by different sources (e.g., organizational vs. individual researcher, source affiliation, source expertise, etc.) may have different impact on dissemination. With the single source in this study, the generalizability of the finding is undermined.

Response 2.7: Whilst we think that the impact of the animation would be generalizable to similar Twitter accounts, we agree that the source is a possible influence upon generalizability and have include this as a limitation: “The animations were disseminated from the personal account. It is possible that the impact of the animations was affected by the source; future work should consider the influence of changing the source, for example, to an organisational account.” (Line 260)

8. As a reader, I would also like to see more discussion of using animation in comparison with other possible media formats to enhance dissemination (e.g., podcast interview, infographics). 

Response 2.8: We did not examine alternative approaches. We have therefore summarised related findings about a range of dissemination methods, including podcasts and infographics. (Line 87, Line 295)

9. The appendix numbering should be checked and fixed. The A, B, C numbering system and the 1, 2, 3 numbering system are used at the same time.

Response 2.9: We apologise for this error and note that we have corrected the numbering.

10. I would suggest deleting appendix B because the study is not a randomized control trial.

Response 2.10: We agree and have removed the checklist

11. Some of the sentences are difficult to understand due to complicated sentence structures. Please edit the manuscript to enhance the readability.

Response 2.11: We agree and have reviewed the manuscript to improve readability.

---

## [Editor Report · Decision Letter 1]

14 Jun 2022

Disseminating implementation science: Describing the impact of animations shared via social media

PONE-D-22-05141R1

Dear My Sykes,

We’re pleased to inform you that your manuscript has been judged scientifically suitable for publication and will be formally accepted for publication once it meets all outstanding technical requirements.

Kind regards,

Christopher Carroll, Ph.D.

Academic Editor

PLOS ONE

Additional Editor Comments (optional):

Thank you for the thorough and careful revisions completed.

Please look again at Box 1/Figure 1 - there seems to be a little confusion here (Box 1 is not a flowchart, just a list; Figure 1 is a flowchart, and represents something different) - should they both be there? Do they need separate labelling?
---

## [Editor Report · Acceptance letter]

16 Jun 2022

PONE-D-22-05141R1 

Disseminating implementation science: Describing the impact of animations shared via social media 

Dear Dr. Sykes:

I'm pleased to inform you that your manuscript has been deemed suitable for publication in PLOS ONE. Congratulations! Your manuscript is now with our production department. 

Kind regards, 

on behalf of

Dr. Christopher Carroll 

Academic Editor

PLOS ONE